# Exploring Different Stakeholder Perspectives on Bilingualism in Autism

**Katie Beatrice Howard** [1,*], **Jenny L. Gibson** [2] **and Napoleon Katsos** [3]

1   School of Education, University of Exeter, St Luke's Campus, Exeter EX2 4TH, UK
2   Faculty of Education, University of Cambridge, Cambridge CB2 1TN, UK; jlg53@cam.ac.uk
3   Department of Theoretical and Applied Linguistics, University of Cambridge, Cambridge CB2 1TN, UK; nk248@cam.ac.uk
*   Correspondence: k.howard2@exeter.ac.uk

**Abstract:** An increasing body of research suggests that bilingualism is possible and perhaps even advantageous for autistic individuals. However, several factors might influence parents' decisions about raising their autistic child bilingually, including national language policies, educational contexts, advice received from key professionals, and the child's individual strengths and needs. Accordingly, there is a clear imperative to understand how the views of different stakeholders converge and diverge when language decisions are made in the context of autism. This paper brings new insights by synthesising the findings of three qualitative studies that used interpretative phenomenological analysis (IPA) to explore the perspectives and experiences of bilingual autistic children (n = 11), parents (n = 16), and educational practitioners (n = 13) of bilingualism in autism in England and Wales. Despite wide variation between and within groups, a striking tension emerged between individuals' beliefs about bilingualism in general, which were positive, and their experiences of bilingualism in autism specifically, which often foregrounded more monolingual approaches. This paper examines this tension, with a particular focus on stakeholders' attitudes towards the feasibility of bilingualism, families' language choices in the context of autism, and how notions of contextual linguistic diversity accentuated differences between England and Wales. We conclude by arguing that greater awareness of both bilingualism and neurodiversity in educational and clinical settings may improve the experiences of bilingual autistic children and, crucially, the language advice families receive.

**Keywords:** bilingualism; autism; interpretative phenomenological analysis

## 1. Introduction

Multilingual parents and caregivers often face challenges in deciding which—and how many—languages to use with their children. Parental motivation and attitudes to multilingualism can play a significant role in determining whether or not families decide to maintain their home language(s) (Hollebeke et al. 2022). However, such choices may also be considered more complex when a child in the family is autistic. Autism is characterised by challenges in social communication and repetitive or restricted patterns of behaviour (APA 2013). However, researchers are increasingly understanding autistic traits as differences rather than evidence of disorder, in line with the neurodiversity paradigm (Pellicano and den Houting 2022) and, to some extent, the social model of disability (Woods 2017). Parallels may then be drawn between bilingualism and autism in that both have historically been viewed through the lens of deficit. Similarly, both are considered heterogenous constructs that exist as spectra, although perhaps it is more helpful to adopt a multidimensional approach that better represents the full range of experiences of bilingualism, autism, and the interaction between them.

A growing body of research suggests that bilingualism is not detrimental to autistic children's cognitive, social, or linguistic development (Reetzke et al. 2015; Siyambalapitiya et al. 2022; Zhou et al. 2019; for a review see Uljarević et al. 2016). In contrast, tentative

findings indicate that bilingual exposure may benefit certain aspects of inhibitory control, cognitive flexibility, and theory of mind among autistic children (Montgomery et al. 2022; Peristeri et al. 2021a, 2021b); these findings appear even in non-clinically diagnosed populations with high autism-like traits (Kašćelan et al. 2019). However, many multilingual families are advised to, or decide to, opt for a more monolingual approach to raising their autistic child(ren) due to concerns that the presence of two or more languages may be confusing (Hampton et al. 2017; Yu 2009). Unsurprisingly, parental uncertainty around language decisions in the context of autism is rife (Davis et al. 2023; Howard et al. 2021).

A wide range of factors influence parents' decisions about bilingualism in autism, including the advice they receive, family context, and their child's developmental profile (Digard et al. 2023; Howard et al. 2021; Sher et al. 2022). As such, evidence from clinical recommendations and existing research suggests that language choices and practices should be made on a case-by-case basis (Hampton et al. 2017; Lim et al. 2018). The issue is particularly pertinent because family language decisions can be the source of much parental stress and anxiety (Sevinç 2022), even though bilingualism may positively influence children's well-being (Müller et al. 2020). Indeed, for those who choose to raise their autistic child in a multilingual environment, benefits may include enriched relationships with family and community members (Digard et al. 2023; Yu 2016), participation in religious and cultural life (Howard et al. 2019a; Jegatheesan 2011; Sher et al. 2022), and, in adulthood, more social and vocational opportunities (Digard et al. 2022). Recent research also suggests that language learning is a particular passion or strength for some autistic adults (Caldwell-Harris 2022).

However, until recently, the voices of bilingual autistic children themselves were conspicuously absent from research into bilingualism in autism, and little is known about their experiences as bilinguals.

Given the significant role teachers play in shaping children's linguistic repertoires (Cunningham 2019) and the fact that schools are important sites for providing families with advice on child development, understanding educators' influence on families' language decisions and practices is also crucial. While schools and teachers may hold positive attitudes about multilingualism, these attitudes do not always translate into meaningful practice, and multilingualism in schools may be more 'tolerated' than actively welcomed (Cunningham and Little 2022). Research suggests teachers are likely to prioritise the development of English over multilingualism (Bailey and Marsden 2017) or may prioritise so-called 'higher status' languages (Amankwah and Howard 2024). Moreover, professionals may see home language maintenance as primarily the responsibility of families rather than schools (Gkaintartzi et al. 2015; Weekly 2020. In the case of decisions about bilingualism in autism, recent studies have found a mixed response to teachers' views; while Howard et al. (2020) found that several educational practitioners had concerns about bilingual autistic children's literacy development in English, Sher et al. (2022) found that most educators in their sample would advise a multilingual approach provided the child had no cognitive difficulties. Despite research suggesting school-based support for bilingual autistic children should be delivered in the home language (Beauchamp and MacLeod 2017), a lack of bilingual staff and bilingual special education services mean that it is not necessarily an autistic child's *capacity* for bilingualism that may prevent them from maintaining their home language but a lack of *opportunity* for bilingualism (Paradis et al. 2018).

*The Current Study*

Heeding Sher et al.'s call (2022) to consider areas of convergence and divergence between different stakeholders' viewpoints, the aim of this paper is to bring together the findings of three qualitative studies (Howard et al. 2019a, 2020, 2021) seeking the experiences of children, parents, and educational practitioners in relation to bilingualism in autism. While the individual studies provide important insights into each group's experiences of bilingualism in autism, only a multi-perspectival account can paint a more

nuanced picture of both individual cases and collective differences between stakeholders, arguably resulting in a more convincing and substantive analysis (Larkin et al. 2019).

In these studies, participants were recruited from two linguistically different settings, adding a further layer of complexity to our analysis: (1) in England, children spoke or had access to language(s) other than English at home and were educated in English; (2) in Wales, children had access to both Welsh and English at home and in school. The context, therefore, mirrors Bialystok's distinction (Bialystok 2018) between 'bilingual education' (as in Wales) and 'the education of bilingual children' (as in England). Exploring differences between these environments enables us to account for the role of contextual linguistic diversity (Wigdorowitz et al. 2022) on experiences of bilingualism in autism. Our analysis was, therefore, guided by the following research question and sub-questions:

(1)  To what extent do the perspectives and experiences of children, educators, and parents converge and diverge when bilingualism meets autism?

    a.  How are perspectives about bilingualism and the feasibility of bilingualism in autism similar and different across participant groups?

    b.  How do lived experiences of bilingualism in autism converge and diverge across the three groups?

    c.  In what ways do participants' accounts differ between England and Wales?

## 2. Methods

### 2.1. Context of the Current Synthesis

The current article brings together the findings of three qualitative studies seeking the perspectives and experiences of children (n = 11) (Howard et al. 2019a), family members (n = 16) (Howard et al. 2021), and educational practitioners (n = 13) (Howard et al. 2020) in relation to bilingualism in autism. The methods for each study are described in depth in the respective studies; therefore, only an overview will be provided here.

The three studies took place in schools in England and Wales, enabling some contextual comparison between bilingual autistic children who were educated bilingually and those who had access to other languages at home. Of the schools in England, three had a percentage of pupils speaking 'English as an additional language' (EAL) below the national average and four had a percentage of EAL pupils above the national average. In Wales, three children attended Welsh-medium schools (in which the language of instruction was Welsh) but came from English-dominant homes, while three children attended English-medium schools but spoke more Welsh at home.

### 2.2. Participants

Participants (n = 40) were selected using purposive sampling, which relies on the researchers' judgement to choose participants who are representative of the population under investigation. Participants were recruited through direct contact with mainstream schools, email bulletins sent by autism organisations, communication with parental support groups, and social media posts. Children and parents were included when the child (1) had been diagnosed with autism in the UK and (2) was exposed to more than one language on a daily basis. Families from a wide range of language backgrounds (Welsh, Bengali, Hindi, Turkish, Spanish, Urdu, Punjabi, Italian, Polish, Gujarati, Lithuanian, French, Arabic) were included in this study. Educators who worked alongside the participating families were then contacted.

Where possible, 'triads' of participants (e.g., a child, a family member, and an educational practitioner) were recruited together. In many cases access to all three was not possible. For example, some parents felt that their child would not be able to meaningfully participate in a verbal interview but still wished to take part themselves, while some schools declined the invitation for staff members to be involved due to time constraints. Demographic information about the participants is found in Table 1.

**Table 1.** Participant information.

| | Child (Gender) | Age | Interview Length (Location) | Language(s) Other than English | School Type | Parent (Gender) | Interview Length (Location) | Language Decision [1] | Practitioner | Interview Length (Location) | Country |
|---|---|---|---|---|---|---|---|---|---|---|---|
| 1 | Male | 6 | - | Welsh | Mainstream Primary (WM [2]) | Mother | 30:50 (school) | Mono | Teacher | 20:51 (school) | Wales |
| 2 | Male | 7 | 20:44 (home) | Bengali Hindi | Mainstream Primary | Mother Father | 35:17 (home) | Mono | - | - | England |
| 3 | Female | 7 | - | Turkish | Autism Unit in Mainstream Primary | Mother | 18:53 (home) | Mono | - | - | England |
| 4 | Male | 8 | 17:23 (home) | Welsh | Mainstream Primary (WM) | Mother | 22:23 (home) | Multi | - | - | Wales |
| 5 | Male | 8 | 23:14 (home) | Spanish | Mainstream Primary | Mother | 27:43 (home) | Mono | SENCO [3] | 19:44 (school) | England |
| 6 | Male | 9 | 14:14 (school) | Hindi | Mainstream Primary | Mother | 28:10 (school) | Multi | Teacher | 15:17 (school) | England |
| 7 | Female | 9 | 9:10 (school) | Urdu Punjabi | Mainstream Primary | Mother | 12:58 (school) | Multi | Teaching assistant | 18:01 (school) | England |
| 8 | Male | 9 | 12:47 (school) | Italian | Mainstream Primary | Mother | 26:59 (school) | Multi | Teacher | 17:04 (school) | England |
| 9 | Male | 9 | 7:55 (school) | Polish | Mainstream Primary | Mother | 16:21 (home) | Multi | Teacher | 18:11 (school) | England |
| 10 | Male | 9 | 13:08 (school) | Welsh | Mainstream Primary (WM) | Mother Grandmother | 35:32 (school) | Multi | SENCO | 9:13 (school) | Wales |
| 11 | Male Male | 9 11 | - | Welsh | Autism unit in EM [4] Mainstream Primary | - | - | - | Teacher SLT [5] Teaching assistant Teaching assistant | 38:34 (school) | Wales |
| 12 | Male | 10 | 16:03 (school) | Hindi Gujarati | Mainstream Primary | Mother | 42:37 (school) | Mono | Teaching assistant | 13:08 (school) | England |
| 13 | Male | 11 | 21:47 (home) | Welsh | Specialist Autism Primary School (EM) | Mother | 26:28 (home) | Multi | - | - | Wales |
| 14 | Male | 12 | 18:05 (home) | Italian | Mainstream Secondary | Mother | 46:04 (home) | Multi | - | - | England |
| 15 | Male | 14 | 19:57(school) | Lithuanian | Mainstream Secondary | - | - | - | SENCO | 39:40 (school) | England |
| 16 | Male Female | 18+ 18+ | - - | French Arabic | Specialist Autism School Mainstream Secondary | Mother | 23:13 (public space) | Mono | - | - | England |

[1] More multilingual approach (multi) or more monolingual approach (mono). [2] WM = Welsh medium. [3] Special Educational Needs Co-Ordinator. [4] EM = English medium. [5] Speech and Language Therapist.

Pseudonyms were attributed to all participants in the original studies. However, in light of researchers' ethical responsibility to ensure that internal confidentiality is upheld and the increased likelihood in a multi-perspectival design that participants might identify themselves or others (Ummel and Achille 2016), no names were used in the current study. Instead, specific quotations were attributed anonymously to 'a child', 'a parent', or 'a practitioner'.

*2.3. Procedure*

Ethical approval was sought from and granted by the School of the Humanities and Social Sciences at the University of Cambridge before the studies began (Case No: 17/136). Semi-structured, phenomenological interviews were conducted by the first author with all participants to understand their experiences in relation to bilingualism in autism. Interviews took place at a location of the participants' choosing, and all three interview schedules were piloted. Additional provisions were made to support the inclusion of children in interviews, including the use of a 'stop/move on' card, an 'emoji' palette as a visual prompt they could point to when faced with difficulty expressing themselves verbally, and a computer-assisted interviewing technique. This involved showing children five images on a computer screen, which pertained to five domains of school experience (language use, socialisation, accomplishment, motivation, and environment) and served as a platform for discussion. Some adults were interviewed together. For example, in the parent group, one father and one grandmother were interviewed together with the two respective mothers, and in the educator group, four practitioners from the same school were interviewed as a group to discuss two children (see Table 1). Two parents were interviewed alongside an interpreter.

*2.4. Data Analysis*

2.4.1. Interpretative Phenomenological Analysis

Interpretative phenomenological analysis is a qualitative research approach developed by Smith et al. (2009), which seeks to describe and interpret individuals' lived experience. Smith (2004) characterises the approach as idiographic, inductive, and interrogative. Intrinsic to IPA is a 'double hermeneutic', whereby the researcher aims to interpret the experience of participants who are themselves actively engaged in a sense-making process (Smith and Osborn 2015). This approach is considered particularly useful within autism research because of its focus on understanding participants' lived experience through their own words. Accurately reflecting the perspectives of autistic individuals is particularly important in light of the 'double empathy problem' (Milton 2012), whereby mutual misunderstanding between non-autistic and autistic people may undermine the authenticity of autism research. Certain features of IPA, including its commitment to an equality of voice and researcher reflexivity, may help to illuminate the experiences of autistic individuals and mitigate the double empathy problem (Howard et al. 2019b).

2.4.2. Multi-Perspectival IPA

A multi-perspectival IPA design was chosen in order to elicit to a more nuanced picture of the experiences of bilingual autistic children. While IPA has traditionally opted for more homogenous samples, a recent trend has emerged towards multi-perspectival approaches that enable researchers to consider 'the relational, intersubjective, and microsocial dimensions of a given phenomenon' (Larkin et al. 2019, p. 183). It could be argued then that the 'double hermeneutic' central to IPA becomes a 'triple hermeneutic' when multi-perspectival designs are employed; the researcher and participant are not only interpreting the participant's own experiences but also seeking to understand the sense-making of others. For example, educators and parents were making sense of their own experiences of bilingualism in autism while simultaneously seeking to understand the children's experiences. Accordingly, the synthesis of viewpoints—not only within but also across participant groups—may bring about a more convincing and cogent analysis than a

single-group design (Larkin et al. 2019). Given the large overall sample of this synthesis (n = 40), there are risks of losing the idiographic nature of IPA with a multi-informant design. As such, attempts have been made to provide examples of individual triads of participants to better understand unique experiences of bilingualism in autism.

Difficulties may also arise when different groups give conflicting advice or opinions (Fletcher-Watson et al. 2019). Within the current sample, it has, therefore, been particularly important to reflect the diversity of views expressed both within and between groups and avoid homogenising a single 'bilingual autistic experience'. Disparities between the amount of information given by different participant groups in the sample also resulted in unequal coverage of viewpoints. Most notably, parents and practitioners reflected more about their attitudes and practices in relation to bilingualism in autism and provided more lengthy accounts than the children themselves. Children may have provided less information for several reasons, including the fact that (1) some were unaware of their autism diagnosis, (2) some were not aware they were bilingual, (3) some had language difficulties that restricted their participation, and (4) it is unreasonable to expect children to articulate detailed opinions in the same way as adults. Where possible, the experiences of the children themselves have been foregrounded to mitigate such disparities.

### 2.4.3. Methods for Cross-Group Analysis

By integrating the three distinct IPA studies, this synthesis draws on 'directly related groups', that is, groups who are 'involved with the same phenomenon, but that are likely to have distinct perspectives on it' (Larkin et al. 2019, p. 187). The cross-case analysis presented here was not based on the frequency of themes across groups but rather themes that are most relevant to answering the research questions relating to perspectives and experiences of bilingualism in autism. Following Larkin et al.'s recommendations, each micro-system, that is, each participant group, was considered individually before moving 'outwards' (Larkin et al. 2019, p. 190) to analyse areas of convergence and divergence across the three groups. Drawing on the strategies outlined by Palmer et al. (2010) for applying IPA to focus group data and recommendations put forward by Larkin et al. (2019), eight steps were taken to arrive at the cross-group analysis. These are presented in Table 2. The analysis was led by the first author and supported by regular discussions within the research team to increase the trustworthiness of the findings.

**Table 2.** Analytical process for multi-perspectival IPA.

| Step | Action |
|---|---|
| 1 | Superordinate and subordinate themes from each participant group were organised into two categories: 'perspectives' or 'experience' (see Table 3) |
| 2 | Patterns were identified between the superordinate and subordinate themes within the 'perspectives' and 'experience' columns, respectively. Two new themes were created for the 'perspectives' category ('attitudes towards bilingualism' and 'feasibility of bilingualism in autism') and four from 'experience' ('children's language use', 'well-being and educational consequences of language choices', 'identifying challenges', and 'improving school experience'). |
| 3 | All transcripts were re-read to ensure that the selected themes were appropriate. This also meant that data not previously presented in the three original studies could be included. |
| 4 | Areas of convergence across all participant groups, first for the 'perspectives' category, then for the 'experience' category, were identified. |
| 5 | Areas of divergence between two or more participant groups were identified, first for the 'perspectives' category, then for the 'experience' category. |
| 6 | Triads of participants (or in some cases, dyads and tetrads) were identified that reflected the specific areas of convergence or divergence. |
| 7 | Areas of convergence and divergence between the two linguistically different settings were noted in light of the above findings. |
| 8 | Findings were evaluated in the wider context of the existing literature in keeping with IPA's interrogative approach. |

**Table 3.** Themes related to 'perspectives' and 'experience'.

| Participant Group | Superordinate Themes Related to 'Perspectives' | Subordinate Themes | Superordinate Themes Related to 'Experience' | Subordinate Themes |
|---|---|---|---|---|
| Children | Identity Formation | Being bilingual | Identity Formation | Developing as learners |
| | | | | Social identity |
| | | | School experience | Learning environments |
| | | | | Well-being |
| Practitioners | Perspectives on bilingualism in autism | Bilingualism for typically developing children vs. bilingualism for autistic children | Perspectives on bilingualism in autism | Consequences for the classroom |
| | | | Creating inclusive learning environments | Identifying barriers to learning |
| | | Concerns about feasibility | | Best practice in the classroom |
| | | | | Whole-school approaches |
| | Perceptions about the value of bilingualism | Impact on communication | Consequences of language choices | Family well-being |
| | | Cultural value | | Children's language use |
| | | Impact on cognition | | Education |
| Parents | Factors influencing language decisions | Feasibility of bilingualism | | Communication with family |
| | | Practical considerations | Factors influencing language decisions | |
| | | The role of English | | |
| | Shifting expectations | Future language learning | | Advice received |
| | | Language choices are not fixed | | |

## 3. Findings and Discussion

This research sought to illuminate the lived experiences of children, parents, and educators in England and Wales when bilingualism meets autism. In particular, we were interested in how perspectives and experiences differed between stakeholders, how this might influence language decisions in multilingual families, and what differences were observed between the two linguistically different contexts. A striking tension emerged in our analysis between individuals' beliefs about bilingualism in general and their experiences of bilingualism in autism in practice. As such, we divide superordinate themes into 'perspectives' and 'experiences' when bilingualism meets autism. The first step in our cross-group analysis involved organising existing themes into these two categories, which are presented in Table 3.

The second step of the analysis involved finding patterns between the superordinate and subordinate themes within the two categories in relation to the research questions. From the 'perspectives' column, two new themes were created: (1) attitudes towards bilingualism and (2) the feasibility of bilingualism in autism. From the 'experiences' category, four themes were developed: (1) children's language use; (2) well-being and educational consequences of language choices; (3) identifying challenges; (4) improving school experiences. These categories, presented in Table 4, will now be discussed with reference to how the accounts of children, practitioners, and parents converge and diverge. Only the first two subthemes of each theme will be discussed in order to answer our research questions in relation to language attitudes and practices.

Before presenting the themes, it is important to note that of the 14 families where parents participated, eight indicated that they had opted for a more multilingual approach to raising their autistic child, while six reported opting for a more monolingual approach (i.e., using mainly English). Three out of the four families interviewed in Wales opted to maintain Welsh, while five out of ten families in England opted to maintain their home language (Hindi, Urdu, Punjabi, Italian, and Polish).

**Table 4.** 'Perspectives' and 'experiences' across groups.

| Superordinate Theme | Subordinate Themes |
|---|---|
| 'Perspectives' of bilingualism in autism | (1) Attitudes towards bilingualism <br> (2) Feasibility of bilingualism in autism |
| 'Experiences' of bilingualism in autism | (1) Children's language use <br> (2) Well-being and educational consequences of language choices <br> (3) Identifying challenges <br> (4) Improving school experiences |

*3.1. Perspectives of Bilingualism in Autism*

3.1.1. Attitudes towards Bilingualism

Children, educational practitioners, and parents tended to hold positive views about bilingualism across the three studies. However, while almost all participants identified benefits of bilingualism, only participants from the parent group noted benefits of bilingualism that were specific to autistic children. Practitioners and children, in contrast, spoke of benefits applicable to the general population without recourse to autism or their specific circumstances. Some practitioners mentioned cognitive, cultural, and vocational advantages to bilingualism, but none identified a benefit of bilingualism in relation to autism. Instead, some were concerned that bilingualism had a negative impact on their bilingual autistic pupil. Similarly, the majority of children discussed advantages of bilingualism in a general sense, often using the second-person pronoun 'you', rather than relating those benefits to their own context; for example, 'you can meet people in other countries', 'you can help people in other languages', and 'it's good to speak Hindi because it's good to pray to God'. Despite reporting some benefits to bilingualism, many children minimised their own linguistic capacity and the intrinsic value of their home language. This is incongruent with the views of their parents, who unanimously commended home language maintenance, even if, as was the case for six families, they had opted for a more monolingual approach.

A possible reason for this disconnect between children's and practitioners' views on the one hand, and parents' views on the other, is that parents were predominantly bilingual themselves. It stands to reason that bilingual individuals will hold more favourable attitudes towards bilingualism than monolinguals (like many of the educators) or emergent bilinguals (like many of the children), as they are able to draw on their personal experiences of bilingualism. This line of argument is bolstered by the manner in which practitioners working in more multilingual educational settings and children whose parents had adopted a more multilingual approach also held more positive views about bilingualism. Greater exposure to multilingualism either in the home (e.g., parents opting for a more multilingual approach) or schools (e.g., bilingual schools or those with a high percentage of EAL students) was indicative of more positive attitudes towards bilingualism. In our analysis, participants in multilingual environments in England often had more in common with participants in Wales than those in England who were based in more monolingual settings. Undoubtedly then, the context in which participants found themselves had a significant impact on their attitudes towards bilingualism, as found in previous research (e.g., Bailey and Marsden 2017). This reinforces the need to account for contextual linguistic diversity (Wigdorowitz et al. 2022) when exploring not only language profiles but also language attitudes towards bilingualism in the context of neurodiversity.

The difference in perspectives on bilingualism between parents, practitioners, and children reveals a wider tension of priorities, as also noted in previous studies (Lee and Oxelson 2006; Zhang and Slaughter-Defoe 2009). Namely, parents may be more inclined to maintain the home language or, in Wales, enrol their child in a bilingual education system, whereas children and practitioners preferred—or at least, prioritised—English. In their reluctance to acknowledge their home language, children gave superior status to English. In this sense, a strong divergence between parents' and children's accounts emerged in that children often reported being less proficient in the home language than their parents had

indicated. For example, one child said of his home language 'I don't speak it that much', while his parent reported it being the primary language used at home. Regardless of actual proficiency, what emerges here is an attitudinal difference. Children may have downplayed their bilingual abilities in order to assimilate to the monolingual norms of classrooms in England, while parents hoped to preserve the home language, in line with findings in Zhang and Slaughter-Defoe (2009). However, children's and parents' accounts converged when it came to the value of bilingualism for communicating with family members (cf. Kwon 2017). This was even true for parents who had opted for a more monolingual approach, many of whom expressed frustrations or concerns that the child may miss out on important familial relationships. Conversely, only one practitioner discussed the relational benefit of bilingualism. In many ways, it is unsurprising that educators would concentrate less on the benefits of bilingualism that affect the familial domain, as their focus is on the child's educational development and they may not view encouraging home language maintenance as part of their role, as has been found in previous studies (Gkaintartzi et al. 2015; Weekly 2020).

Indeed, like many of the children, some practitioners were also keen to stress the importance of English over the home language. This may have related to the fact that they associated English proficiency with academic success, as is the case in previous research (Bailey and Marsden 2017). Such a finding could also be partially attributed to a trend in which practitioners focused on the child's identity as an autistic learner rather than a multilingual one. On the one hand, this trend may have been specific to the sample, as very few of the children were born outside the UK; therefore, their levels of English proficiency were high. On the other hand, the challenges faced by autistic pupils described in the existing literature tend to be emphasised more than those faced by bilingual, or EAL, pupils; therefore, it is possible that practitioners foregrounded the child's autistic identity at the expense of their linguistic one. This was less prevalent in the accounts of practitioners in Wales, perhaps because their pedagogy was often centred around the teaching and learning of bilingual pupils.

One triad of stakeholders models in microcosm the differing perspectives on bilingualism between the three groups. Consistent with the notion of children minimising their home language proficiency, the child was reluctant to identify as a multilingual in his affirmation that 'I just know English and a tiny bit of Spanish'. He then highlights that bilingualism is beneficial because 'when you're older you can speak to people that are in those languages'. His use of 'you' and 'when you're older' might establish some distance between his own linguistic practices and his perception of the ideal multilingual individual. Meanwhile, his mother highlighted her child's daily exposure to Spanish, saying he is 'constantly hearing Spanish in the house' and describing his Spanish as 'inbuilt'. Although she conceded that his understanding of Spanish was greater than his production, their accounts are partially conflicting. Despite opting for a more monolingual approach, the mother highlighted several benefits of bilingualism for her son, including better communication with immediate family members, potential academic gains (such as increased confidence), and further opportunities for social interaction. Like many of the triads of participants, these sentiments were not shared by the educational practitioner working with the child. Instead, the practitioner believed that bilingualism was problematic, i.e., 'having English as an Additional language impacts on his processing, therefore it would be a negative for him'. While the mother was focused on the familial aspects of home language maintenance, the educator's perspective was shaped by her educational role in his life, which led to her belief that bilingualism was more of a hindrance than a help. The conflicting priorities and perspectives in this triad are symptomatic of the divergence present across different participant groups in this study.

### 3.1.2. Feasibility of Bilingualism in Autism

Perspectives on the feasibility of bilingualism in autism also diverged both between and within participant groups. For example, within the parents' group, families who



opted for a more multilingual approach believed that being bilingual may bring some advantages to their child's autism, while families who chose a more monolingual approach to raising their child reported either no effect or concerns that bilingualism may be inimical to their child's development. Unlike Sher et al.'s (2022) findings, in which both practitioners and families held a strong preference for the maintenance of bilingualism in autism, and Davis et al. (2023), where speech and language therapists tended to encourage multilingual approaches in autism, our findings suggested that educators were more likely to have concerns about the feasibility of bilingualism in autism, even though such concerns are incongruent with research suggesting that there is no detrimental effect of bilingualism on autistic children (Montgomery et al. 2022; Uljarević et al. 2016). That said, some educators argued that in the right conditions and with the right support, it was possible for an autistic child to develop more than one language.

Several parents and practitioners shared the view that the feasibility of bilingualism depended on the autistic presentation and profile of the individual child. Accordingly, they believed that languages decisions should be made based on the extent to which the child could communicate their basic needs. For instance, one educator argued that the child's ability to communicate their fundamental needs was more crucial than developing their bilingual proficiency. This view was corroborated by a parent, who stated that she would not have pursued bilingualism if her son had not been able to express his basic needs. Families' language choices were often made, whether consciously or not, along these lines. Similarly, several parents reported that they used their own language with their child because it came more naturally, while others chose a more monolingual approach because using two languages came less naturally to the child. Some participants in both groups expressed apprehension that the child was becoming confused by the presence of two languages, in keeping with the findings from Hampton et al. (2017), although this was a more prevalent belief among practitioners. Across the two groups, participants felt that bilingualism was possible for some, but not all, autistic children.

Some consensus was reached between parents and practitioners about the feasibility of bilingualism in specific cases. For example, in one case, while both the parent and practitioner strongly advocated for bilingualism, they both believed that bilingualism was not suitable for the specific child under discussion, stating that bilingual exposure was potentially 'holding him back' and 'making a difference on his confidence'. As a result, the parent had decided to move her son from a Welsh-medium to an English-medium primary school the following academic year based on advice from the practitioner. In this instance, the Welsh-medium school had made significant efforts to support the child's emergent bilingualism, but both the family and school agreed that a monolingual environment was more appropriate. This case illustrates that parents' and practitioners' beliefs about bilingualism may not be the decisive factor when choosing which, and how many, languages to use with an autistic child.

There were also examples of divergence between individual triads. For example, one practitioner argued that bilingualism was having a detrimental impact on the student's written and spoken English and subsequently suggested that his parents could speak more English at home to encourage his English proficiency. This advice, though well intentioned, may not only be impractical to implement but also unjust (Gréaux et al. 2020; Sher et al. 2022). Asking the family to model their non-native language may have unintended consequences for the child's development of English (Davis et al. 2023) and wider family well-being (Müller et al. 2020) and was particularly problematic in this case, as the parent relied on an interpreter to complete the interview. This case demonstrates that, for some families, maintaining the home language was more of a necessity than a choice, parents' own language proficiency can play a significant role in language choices (cf. Drysdale et al. 2015), and parents and educators may have competing priorities when it comes to home language maintenance.

### 3.2. Experiences of Bilingualism in Autism

3.2.1. Children's Language Use

Just as participants had varying perspectives about bilingualism (in autism and more broadly), so too did their experiences of language inevitably concur in some areas and differ in others. One central area of convergence between participants' accounts relates to the way in which children compartmentalised their languages between different contexts, most commonly between home and school. Parents in Wales particularly emphasised this point, stating that their child considered Welsh to be the 'language of school', and, in most cases, English was 'the language for home'. A similar trend was noted in England; children reported preferring to use English at school, in line with previous findings (Liu and Evans 2016), and did not wish to conflate their linguistic spaces. Some children were surprised even to be asked if they used languages other than English in school. When asked 'when do you speak Hindi?', one child responded, 'when I am in India', which shows a clear compartmentalisation of language use despite his mother reporting that Hindi was the main language used at home. Similarly, practitioners commented that they rarely heard the child use their home language in school. For example, one practitioner described a child's resistance to speaking their home language with a bilingual teaching assistant during sessions designed to support home language maintenance, while other practitioners who had concerns about bilingualism in autism reported that they seldom heard the child using their home language in school. These trends may explain why less emphasis seems to be placed on the child's multilingual identity in the school environment. Perhaps if children in the sample had been new to English, these reflections and their compartmentalisation of language may have been different.

The accounts of all participant groups also converged to a large extent regarding children's use of English. Firstly, most children stated that they felt more comfortable using English than their home language, or Welsh for those with English as a first language, which was corroborated by practitioners and parents. In like manner, a common trend running through the three participant groups was that the child could understand the home language (or Welsh) but lacked proficiency, or at least confidence, in speaking it. Some parents decided that developing the child's receptive knowledge of the home language was more realistic than expecting them to become fluent speakers. This chimes with children's own reported lack of confidence with using their home language, as exemplified by the repetition of 'just' and 'only' in reference to speaking their home language. As expected, this was particularly the case for children whose families had adopted a more monolingual approach, such as one child, who affirmed 'I don't speak other languages'. What emerged was the idea that children may not identify as bilinguals if they cannot fluently speak the home language, even if they understand the language well. Such a finding is common within the literature on the linguistic identity of multilingual learners, many of whom are reluctant to claim multilingual competence or identities (Dressler 2014).

Some participants in the parent and practitioner groups also discussed the idea of the child's linguistic and developmental trajectory evolving over time. A sense emerged that while bilingual development was possible for autistic children, in certain cases, it might take longer than for non-autistic children. For example, some participants indicated that there was a different balance of challenges and benefits of bilingualism depending on the child's age; that is to say, there may be more challenges and fewer evident benefits in the earlier stages of development, but the balance may tip the other way as the child grows. This was typified by one educator's belief that 'at the moment it makes it more challenging, but from my understanding, I think when he's older it will be beneficial'. As children in the sample were different ages, it was relatively easy to identify examples of this phenomenon. In Wales, for instance, a practitioner and parent dyad believed that bilingualism was not feasible for the child at age 6. Interestingly, another dyad described having similar feelings in the first few years of primary school and the mother considered adopting a more monolingual approach by moving the child to an English-medium school. However, at the time of the interview, when the child was 9, the accounts of both the parent

and practitioner converged in their belief that the child was now flourishing in a bilingual educational system. This suggests that, in some cases, it may be helpful to persist with a bilingual environment for as long as possible to ensure that the child has sufficient time, exposure, and opportunity to develop bilingual competence. In this vein, some parents' and practitioners' accounts indicated that the child's capacity for bilingualism may change over time. Especially among parents of younger children in the sample, there was a sense that bilingualism may be possible in the future; one parent noted 'when he will be ready, we'll start'. This reinforces the idea that one-off universal advice encouraging multilingual families to adopt a more monolingual approach could have profound and unintended consequences for the child, as well as for their immediate and extended family, as it does not account for developmental changes over time.

### 3.2.2. Well-Being and Educational Consequences of Language Choices

Families' language choices had two main types of consequences across participant groups: those related to well-being and those related to education. Firstly, language choices had both positive and negative implications for well-being. Families who opted for a more multilingual approach tended to report more positive experiences related to well-being. Most notably, children were able to maintain communication with family members; both parents and children highlighted that being able to speak the home language was essential for relationships with immediate and extended family. While only one practitioner highlighted this communicative advantage, another added that for her two autistic pupils, 'Welsh is their home language so they're happy', inferring a link between language maintenance and well-being that is beginning to be established in research (De Houwer 2015; Müller et al. 2020). Other parents indicated that being able to speak 'our language' had positive effects on parental well-being too, which, in turn, may positively influence parent–child relations. Research employing longitudinal designs is needed in order to further investigate the relationship between familial well-being and multilingualism in autistic and non-autistic children alike.

Parents who opted for a more monolingual approach reported some negative effects on well-being, both for the child and for themselves as parents. Some parents reported that their child could become distressed from either their code-switching or using a particular language in the 'wrong' context. This finding converges with the children's strong desire to compartmentalise their languages and demonstrates that regardless of parental language choices, language practices may have a significant impact—for better or worse—on autistic children's well-being. Beyond the child's own language proficiency, opting for monolingualism engendered some negative consequences for parents and siblings. Some parents described their guilt or sadness at not being able to share their language with their child. Such findings resonate with Sher et al. (2022), who found that 'forced monolingualism' had resulted in a sense of alienation and difference among multilingual families with an autistic child. Others mentioned the impact of opting for a more monolingual approach on the child's sibling(s); two parents, for example, mentioned that a monolingual approach for their autistic child resulted in a monolingual approach for the whole family. Accordingly, siblings also missed out on the opportunity to learn and maintain the home language.

Regarding the educational consequences of families' language choices, difficulties with literacy emerged as one of the biggest areas of convergence among the three participant groups. This was true regardless of whether parents had opted for a more multilingual or monolingual approach. Some practitioners attributed children's challenges with literacy to the child's bilingualism or code-switching practices at home, while parents described difficulties with reading comprehension and writing but did not speculate on possible causes. Children also highlighted that literacy-based subjects were among their least favourite or most difficult subjects. This finding was exemplified by one triad in which all three participants noted difficulties with literacy, both in the home language (Polish) and English, although only the educator suggested that bilingualism may be exacerbating such difficulties.

Practitioners highlighted the challenges they faced in distinguishing the causes of children's difficulties (autism, bilingualism or a combination), which inevitably resulted in bilingual autistic children's learning needs being identified later than those of their monolingual autistic peers. This was more common in England, particularly in schools with a higher percentage of EAL pupils, than in Wales. Only one parent questioned whether bilingual children may be more susceptible to being diagnosed with autism later than their monolingual peers. She expressed frustrations that staff at her son's nursery had ascribed his challenges with communication to his exposure to his home language rather than underlying neurodevelopmental differences. This experience of developmental differences being attributed to bilingualism is consistent with previous research that reports delays and mistakes in diagnoses for children from linguistically and culturally diverse backgrounds (Yamasaki and Luk 2018).

Social difficulties experienced by some of the children in the school setting may also be linked to their linguistic environments. For example, it is possible that difficulties acquiring the language of instruction (i.e., English or Welsh) may negatively impact children's social interaction with peers. For example, one practitioner in a bilingual educational setting noted how 'when it comes to playing with the children in the classroom there is no word of Welsh really', which demonstrates how language barriers may have contributed to the child's social challenges. This experience may resonate with many other bilingual autistic children, particularly those who are new to the language of instruction. If the child's main opportunity for developing proficiency in the language of instruction is in school (as parents speak a different language at home), then challenges in social interaction will affect both their social and bilingual development. This is especially important as exposure to and experiences of multilingualism may open doors for positive social experiences among autistic adults (Digard et al. 2020).

Another educational consequence of language choices, specific to children in Wales, concerned school placements. Three out of five children in Wales had moved, or were due to move, schools, either from a Welsh-medium (WM) to an English-medium (EM) school or vice-versa. Their parents had to make difficult choices about whether to keep their child in a WM mainstream school without the specialist support they needed or send their child to an EM specialist school without access to a bilingual education. This difficult choice came about due to a lack of specialist autism schools educating through the medium of Welsh. To exemplify this point, one parent wanted her son to attend a more specialist school, given that 'school is where he has the most difficulty' and, more strikingly still, 'he will never achieve his potential in school because the environment is so set up against him'. However, she was also keen for him to maintain Welsh, given that she believed he had an aptitude for languages and in light of the cultural and cognitive benefits of bilingualism that she outlined.

*3.3. Contextual Linguistic Diversity*

In order to better understand how bilingualism impacts autistic children, it is crucial to attend to contextual linguistic diversity (Wigdorowitz et al. 2022) by exploring the sociolinguistic context in which autistic children grow up. The most significant divergence identified in our analysis was neither between or within the three participant groups but between the experiences of those in Wales and those in England. While children in Wales were educated in a bilingual education system and most of their parents were native English speakers, children in England were educated in a monolingual education system and their parents were not native speakers of English. Unsurprisingly, therefore, across participant groups, attitudes towards bilingualism (both in general and in autism specifically) were more positive in Wales than in England. This may have been the result, in part, of both the linguistic profile of the interviewees and the high status of the Welsh language in Wales (Hodges 2012). In Wales, parents had made a conscious decision to pursue bilingualism at the point of choosing a school for their child, whereas parents in England carried sole responsibility for the transference of the home language to their

children. This was particularly challenging for those who were advised to take a more monolingual (i.e., 'English only') approach and may reflect the finding that proportionally more families in Wales opted for a bilingual approach than those in England.

A further distinction between the two groups was that in Wales, children were different to their peers in that they were autistic, but they were not *linguistically* different to their peers, unlike the children in England. Learners in Wales did not experience the 'double difference' faced by some participants in England, and, therefore, more attention was given to alleviating any challenges associated with their autism. Unlike in England, bilingualism was not viewed as a barrier to the child's academic progress, except in one case. Moreover, the language of instruction seemed more fluid in Wales than the strict linguistic parameters set in England, whereby an 'English only' environment was the firm expectation. Although children in both England and Wales reported compartmentalising their languages, practitioners in English-medium schools in Wales still used and encouraged incidental Welsh; a more fluid approach to linguistic practices in the classroom also suggests that the promotion, rather than separation, of different languages is likely to lead to greater familiarity with bilingualism and, accordingly, greater acceptance. In England, in contrast, a distinction emerged whereby children and practitioners in more multilingual educational settings had far more favourable attitudes towards bilingualism (although not bilingualism in autism) than those in more monolingual environments. Nevertheless, educators were less confident than colleagues in Wales about supporting bilingual children, which is reflected in one parent's observation that 'there was no encouragement to learn another language. And I think that is just nationwide'.

Unlike families in England, no parent in Wales reported receiving advice about bilingualism when their child was diagnosed with autism. This finding reflects a concerning trend, also identified by Roberts (2017), that bilingual support and provision for children with additional learning needs in Wales is insufficient, although the introduction of the 'Additional Learning Needs Transformation programme' has highlighted the need for better bilingual provision (Welsh Government 2020). While it is possible that some children will be diagnosed with autism once decisions about the language of instruction have already been made, greater support and advice should be made available for parents of children diagnosed before they start school in Wales, so that families can make informed decisions about the suitability of an English-medium or a Welsh-medium environment. Moreover, some parents in Wales believed that their children did not have access to appropriate educational services due to a lack of Welsh-medium specialist autism schools. As such, some had to choose between the child's bilingual development (in a Welsh-medium school) and their academic or learning needs (in a monolingual specialist school). Such challenges could also apply to international settings where bilingual education is provided.

## 4. Conclusions

This study explored different stakeholders' experiences of bilingualism in autism. The findings indicate that while many participants considered bilingualism to be beneficial in general, such beliefs did not always readily apply to autistic children. This disconnect between positive beliefs about bilingualism in general and apprehension about bilingualism for autistic children was present across all three participant groups: children stated that bilingualism was helpful for others but minimised the importance of their home language in their own lives; educational practitioners raised concerns about the impact of bilingualism on the literacy and language development of their autistic learners, and nearly half of parents in this study had made the difficult decision to limit the use of their home language in favour of English. Nevertheless, participants' perspectives and experiences also varied significantly depending on their context. For example, those in more multilingual environments tended to be more in favour of a more multilingual approach, as did those in Welsh-medium settings in Wales. This indicates the need for an approach that accounts for contextual linguistic diversity in order to better understand bilingualism in autism.

Our analysis was strengthened by its use of a multi-perspectival IPA approach to better understand how different stakeholders view and experience bilingualism in autism. However, some issues arise when adopting this kind of approach. Firstly, there was considerable overlap between participants' perspectives and their experiences, which makes distinguishing between the two somewhat problematic. Secondly, there was a risk of homogenising each group's perspectives and experiences; this was mitigated through the use of 'phenomenologically-informed case studies' to show the unique experiences of individuals and individual triads and uphold the idiographic commitment of IPA. Thirdly, and crucially, there is a danger in multi-perspectival designs that one group's experience will be given pre-eminence over another's. Attempts were also made to ensure that children's voices were given equal status and coverage (Greene and Hogan 2005), despite the fact that the interviews with children yielded less data in terms of both frequency and detail than interviews with adult participants. Future research designs should not only consider how to ensure an equality of coverage between participant groups but also how participatory approaches might improve the foregrounding of autistic experiences (Fletcher-Watson et al. 2019).

This study has varied and significant implications for the lives of bilingual autistic children. The disconnect between some practitioners' concerns about bilingualism in autism and the practical realities and linguistic proficiency of different families raises important questions about the extent to which practitioners should and do influence family language practices. It is clear that greater support and more research-informed advice needs to be provided to multilingual families with an autistic child. Such support could be achieved through (1) increasing awareness in schools that it is possible for autistic individuals to grow up bilingually; (2) forging stronger family–practitioner partnerships so that the wider family's circumstances can be more deeply embedded into the advice and support given; (3) providing tailored, routine advice and resources to families adapted to the child's evolving developmental trajectory (Digard et al. 2023); (4) ensuring that suitable bilingual specialist provision is available in bilingual contexts to prevent parents from having to choose between monolingual specialist settings or mainstream bilingual settings (Paradis et al. 2018), as was the case in Wales; and (5) promoting multilingualism and neurodiversity in schools to raise awareness that difference should not be equated with deficit.

**Author Contributions:** Conceptualization, K.B.H., J.L.G. and N.K.; methodology, K.B.H.; formal analysis, K.B.H., J.L.G., and N.K.; investigation, K.B.H.; resources, K.B.H., J.L.G. and N.K.; writing—original draft preparation, K.B.H.; writing—review and editing, J.L.G. and N.K.; supervision, J.L.G. and N.K.; project administration, K.B.H.; funding acquisition, J.L.G. and N.K. All authors have read and agreed to the published version of the manuscript.

**Funding:** K.B.H. was funded by the University of Cambridge in association with the AHRC-funded project, 'Multilingualism, Empowering Individuals, Transforming Societies' (MEITS). J.L.G. and N.K. were supported by the UK Arts and Humanities Research Council grant 'Multilingualism Empowering Individuals, Transforming Societies' (MEITS) AH/N004671/1.

**Institutional Review Board Statement:** This study was conducted in accordance with the Declaration of Helsinki and approved by School of Humanities and Social Sciences Ethics Committee at the University of Cambridge (Case No: 17/136).

**Informed Consent Statement:** Informed consent was obtained from all subjects involved in the study.

**Data Availability Statement:** Data is unavailable due to ethical restrictions.

**Acknowledgments:** We wish to thank all those who participated in the original studies. We would also like to thank the MEITS project (Multilingualism: Empowering Individuals, Transforming Societies) for not only providing funding for this work but also a stimulating and collaborative academic community for this project to grow. Finally, we would like to thank the reviewers for their constructive comments.

**Conflicts of Interest:** The authors declare no conflicts of interest.

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
