# Peer review of "Exploring Different Stakeholder Perspectives on Bilingualism in Autism"

_languages, doi:10.3390/languages9020066_

Round 1
Reviewer 1 Report
Comments and Suggestions for Authors
I commend the authors for the introduction where the background and rationale for this manuscript are carefully explained. The only red flag here is that three published studies are being integrated to extract insights. A high bar must be set for re-publishing already published data. I feel the author have met that bar. By integrating the three studies, 2 useful comparisons are possible. One is the contrast between a school setting which assumes bilingualism (Wales) vs. a region with the traditional distinction between majority language (the socially powerful language, English) and minority languages (which may be less valued languages of immigrant families). The second contrast is between the three interviewed groups (teachers, parents and autistic children).
The insights of integrating the three studies are needed. It's a lot to ask readers to consult three separate manuscripts and then integrate the findings on their own. The authors are filling an important gap in the literature.
I suggest an addition to the discussion. Autistic adults and often youth may have a special interest in language and languages. This includes learning new languages or learning their history and sociology. This was documented in a recent manuscript:
Caldwell-Harris, C. L. (2022). Passionate about languages, but listening and speaking–¡ Ay, Caramba! Autistic adults discuss foreign language learning. Journal of Multilingual and Multicultural Development, 1-16.
The idea that some autistic people have a special interest and/or proclivity for learning languages is a perspective that supports encouraging bilingualism in autistic children. Stakeholders could read about autistic adults' passion for language learning to get inspired to consider wider support for bilingualism.
I am impressed with some of the themes extracted from interviews. Children minimized their capacity for language and relevance of bilingualism. This is incongruent with their parents. This suggests that children gain ideas from the majority culture about the lower value of minority languages. The children also may have internalized the idea that autistic children have reduced capacity.
Educational practitioners need to be informed about the benefits for children's cognitive and social development of bilingualism.
The following paragraphs stood out for me in their relevance.
Several parents and practitioners shared the view that the feasibility of bilingualism depended on the autistic presentation and profile of the individual child. ...Children reported preferring to use English at school in line with previous findings (Liu & Evans, 2016) and did not wish to conflate their linguistic spaces. Some children were surprised even to be asked if they used languages other than English in school. Similarly, practitioners commented that they rarely heard the child use their home language in school.
Some parents reported that their child could become distressed from either their code-switching or using a particular language in the ‘wrong’ context. This finding converges with the children’s strong desire to compartmentalise their languages, and demonstrates that regardless of parental language choices, language practices may have a significant impact – for better or worse – on autistic children’s well-being. ==> Is this a novel finding?
Important:
Others mentioned the impact of opting for a more monolingual approach on other siblings; two parents, for example, mentioned that a monolingual approach for their autistic child resulted in a monolingual approach for the whole family. Accordingly, siblings also missed out on the 418 opportunity to learn and maintain the home language.
Only one parent questioned whether bilingual children may be more susceptible to being diagnosed with autism later than their monolingual peers. She expressed frustrations that staff at 430 her son’s nursery had ascribed his challenges with communication to his exposure to his home language, rather than 431 underlying neurodevelopmental differences. This experience of developmental differences being attributed to bilingualism is consistent with previous research that reports delays – and mistakes – in diagnoses for children from linguistically and culturally diverse backgrounds
Important
Unlike families in England, no parent in Wales reported receiving advice about bilingualism when their child 484 was diagnosed with autism.
Author Response
Thank you for your clear and insightful observations about the paper. We noted only a couple of changes or additions to the paper which we address below:
- I suggest an addition to the discussion. Autistic adults and often youth may have a special interest in language and languages. This includes learning new languages or learning their history and sociology. This was documented in a recent manuscript:
Caldwell-Harris, C. L. (2022). Passionate about languages, but listening and speaking–¡ Ay, Caramba! Autistic adults discuss foreign language learning. Journal of Multilingual and Multicultural Development, 1-16.
The idea that some autistic people have a special interest and/or proclivity for learning languages is a perspective that supports encouraging bilingualism in autistic children. Stakeholders could read about autistic adults' passion for language learning to get inspired to consider wider support for bilingualism.
Thank you for this useful suggestion. We have added in a reference to the Caldwell-Harris (2022) paper in the introduction to show that for some autistic individuals language learning may be of particular interest or a particular skill:
Recent research also suggests that language learning is a particular passion or strength for some autistic adults (Caldwell-Harris, 2022). (lines 60-61)
- Some parents reported that their child could become distressed from either their code-switching or using a particular language in the ‘wrong’ context. This finding converges with the children’s strong desire to compartmentalise their languages, and demonstrates that regardless of parental language choices, language practices may have a significant impact – for better or worse – on autistic children’s well-being. ==> Is this a novel finding?
This has been discussed briefly in other papers (e.g. Sher et al., 2022), however, the notion of language practices and decisions having different effects on autistic children’s wellbeing depending on the context (e.g., consequences at home and at school, more monolingual vs more multilingual environments, etc.) appears to be a novel finding and is drawn out here by the use of different stakeholders’ perspectives (as discussed on lines 498-508).
Reviewer 2 Report
Comments and Suggestions for Authors
In their manuscript, “Exploring different stakeholder perspectives on bilingualism in autism,” the authors provide a unique multi-perspectival analysis of the perspectives and experiences of autistic children, caregivers, and educational practitioners about bilingualism in autism. Key strengths of the paper include the use of triads, when available, to examine the specific convergence and divergence of perspectives among an autistic child, their own parent, and an educator who works with that child. The authors also used creative means to make it possible to interview children. The comparison between England and Wales also provides an interesting perspective on the role of contextual linguistic diversity. The authors conclude with concrete calls for action to improve support for bilingual autistic children and their families. I have a few questions and recommendations that may help to strengthen the impact of this important paper.
1. Framing: In addition to referencing the neurodiversity paradigm, it may also be relevant to discuss the social model of disability. Although autistic traits may not be evidence of a disorder, it is still the case that autistic individuals can experience disability in a society that has not been designed for them.
2. Purposive sampling: What were some of the criteria considered by the researchers when choosing participants?
3. Communication modalities of child participants: The table describing participants is very helpful. I found myself also wondering about the children’s modes of communication. Would this information be available to include? (e.g., AAC, spoken language in English, Welsh+, Hindi, etc)
4. Details of methods: As I reader, I would have liked to see the interview guides for each group of stakeholders in an Appendix and to hear more about efforts to ensure trustworthiness (e.g., member checking), especially for the data gathered from autistic children and caregivers. However, I defer to the editor on this recommendation, as I imagine that these details may have been available in the original publications in which qualitative findings from each group are reported.
5. Team/individual analysis: Were the analyses completed by a single author or by a team? If there was a team, the roles of different team members and individual coding vs. discussion could be explained when describing the steps of analysis.
6. Centering children’s voices: While the authors note that the children’s interviews yielded less data and discuss efforts to elevate their voices, I still would have liked to hear more from the children in the subthemes to which they did contribute, such as attitudes toward bilingualism and children’s language use. Would it be possible to include more quotes from the children such as that included on line 290?
7. Triads: To highlight how the multi-perspectival design contributes new information beyond the original studies of each group separately, it might have been helpful to include more quotes from triad sets and/or a summary table demonstrating key areas of convergence and divergence across groups.
Thank you for the opportunity to review this manuscript.
Author Response
Thank you for your very useful comments, suggestions and questions in relation to this paper. We have made a number of changes to the manuscript based on your review, which we outline below.
- Framing: In addition to referencing the neurodiversity paradigm, it may also be relevant to discuss the social model of disability. Although autistic traits may not be evidence of a disorder, it is still the case that autistic individuals can experience disability in a society that has not been designed for them.
We have added a reference to the social model of disability alongside the point about the neurodiversity paradigm to show that both have informed the way in which autism is perceived and researched (lines 36-37).
- Purposive sampling: What were some of the criteria considered by the researchers when choosing participants?
Alongside the inclusion criteria mentioned on lines 119-120, the research team was keen for the sample to represent bilingual autistic children without homogenising either the ‘bilingual’ or ‘autistic’ experience. As such, children from a variety of language backgrounds and with a range of communication profiles (e.g. including parents of children with little verbal communication) were selected. This is outlined on lines 178-179 and we have now added a sentence to outline the variety of languages backgrounds of participants (lines 120-122).
- Communication modalities of child participants: The table describing participants is very helpful. I found myself also wondering about the children’s modes of communication. Would this information be available to include? (e.g., AAC, spoken language in English, Welsh+, Hindi, etc)
This is a very useful suggestion, however, as a language background questionnaire was not used in the studies, we do not have full access to this information about participants and therefore would be able to provide communication modalities for some but not all participants. However, as outlined above, we have now added in a sentence to show the variety of participants’ language backgrounds (lines 120-122).
- Details of methods: As I reader, I would have liked to see the interview guides for each group of stakeholders in an Appendix and to hear more about efforts to ensure trustworthiness (e.g., member checking), especially for the data gathered from autistic children and caregivers. However, I defer to the editor on this recommendation, as I imagine that these details may have been available in the original publications in which qualitative findings from each group are reported.
While we would like to provide the interview schedules, this is not possible due to copyright regulations from the original publications. These can, however, be accessed in the first author’s thesis, available here: https://www.repository.cam.ac.uk/items/a480d3bd-5f1f-4ee5-ab51-7cadf1af0c0e
- Team/individual analysis: Were the analyses completed by a single author or by a team? If there was a team, the roles of different team members and individual coding vs. discussion could be explained when describing the steps of analysis.
We have included some clarification about this in the ‘analysis’ section on lines 197-198 where we note: ‘The analysis was led by the first author and supported by regular discussions among the research team to increase the trustworthiness and transparency of the findings.’
- Centering children’s voices: While the authors note that the children’s interviews yielded less data and discuss efforts to elevate their voices, I still would have liked to hear more from the children in the subthemes to which they did contribute, such as attitudes toward bilingualism and children’s language use. Would it be possible to include more quotes from the children such as that included on line 290?
This is a really important point, particularly as we state the importance of centring children’s voices. We have therefore sought to rectify this by adding 5 additional quotations from children, as follows:
- ‘Similarly, the majority of children discussed advantages of bilingualism in a general sense, often using the second-person pronoun ‘you’, rather than relating those benefits to their own context; for example, ‘you can meet people in other countries’, ‘you can help people in other languages’ and ‘it’s good to speak Hindi because it’s good to pray to God’.’ (lines 236-239)
- ‘For example, one child said of his home language ‘I don’t speak it that much’, while his parent reported it being the primary language used at home.’ (lines 263-264)
- ‘When asked ‘when do you speak Hindi?’, one child responded, ‘when I am in India’, which shows a clear compartmentalisation of language use despite his mother reporting that Hindi was the main language used at home.’ (lines 354-356)
- Triads: To highlight how the multi-perspectival design contributes new information beyond the original studies of each group separately, it might have been helpful to include more quotes from triad sets and/or a summary table demonstrating key areas of convergence and divergence across groups.
We agree that it is essential to the aims of this paper to showcase how different stakeholders within dyads/triads had different perspectives on bilingual language use in autism. We have tried to ensure that each section provides specific examples of individual circumstances, which is also important to the idiographic nature of IPA. Some of the examples in point 6 above also help to demonstrate differing perspectives, and we have added a further example of the differing responses within triads on lines 424-426, where we state:
-
- ‘This finding was exemplified by one triad in which all three participants noted difficulties with literacy, both in the home language (Polish) and English, although only the educator suggested that bilingualism may be exacerbating such difficulties.’